# An Assessment of Knowledge, Attitudes and Practices on Pureed Diet Preparation (KAP DYS-PUREE) among Food Handlers in Malaysian Hospitals for Dysphagia Management

**DOI:** 10.3390/healthcare11142026

**Published:** 2023-07-14

**Authors:** Aizul Azri Azizan, Suzana Shahar, Zahara Abdul Manaf, Hasnah Haron, Nurul Fatin Malek Rivan, Nurul Huda Razalli

**Affiliations:** 1Faculty of Food Science and Nutrition, Universiti Malaysia Sabah, Jalan UMS, Kota Kinabalu 88400, Malaysia; aizul83@ums.edu.my; 2Nutrition Program and Centre for Healthy Aging and Wellness, Faculty of Health Sciences, Universiti Kebangsaan Malaysia, Jalan Raja Muda Aziz, Kuala Lumpur 50300, Malaysia; hasnaharon@ukm.edu.my (H.H.); fatinmalek93@gmail.com (N.F.M.R.); 3Dietetic Program and Centre for Healthy Aging and Wellness, Faculty of Health Sciences, Universiti Kebangsaan Malaysia, Jalan Raja Muda Aziz, Kuala Lumpur 50300, Malaysia; suzana.shahar@ukm.edu.my (S.S.); zaharamanaf@ukm.edu.my (Z.A.M.)

**Keywords:** dysphagia, knowledge, attitude, practice, pureed diet

## Abstract

This study aims to assess the knowledge, attitude, and practice towards pureed diet preparation among food handlers using a newly developed questionnaire for dysphagia management. A descriptive cross-sectional design study with purposive sampling was conducted in three government hospitals in the Klang Valley, Malaysia. A newly developed questionnaire, comprised of 40 quantitative items, was used and distributed to 161 food handlers from three hospitals who were directly involved in preparing pureed diets. The results demonstrated a low to moderate knowledge score among food handlers (57.54 ± 12.33), with scoring at 1.95% (very low), 28.6% (low), and 68.3% (moderate). Only 1.2% scored well in the knowledge section on pureed diet preparation. The attitude among food handlers showed that they were referred to the right source of reference before preparing the pureed diet (3.97 ± 1.35). The findings also clearly showed that the practice of using the right equipment (4.41 ± 1.19) is essential for pureed diet preparation. In conclusion, this study serves as a prognosis for future improvement in knowledge, attitude, and practice among food handlers toward pureed diet preparation. Knowledge among food handlers needs to be enhanced, and a comprehensive guideline and reference module will aid in refining dysphagia management, specifically in food preparation by food handlers.

## 1. Introduction

Globally, the number of patients with dysphagia is increasing due to various factors. The prevalence of dysphagia, which is mainly characterized by difficulty swallowing, may vary across countries, age groups, and diseases that contribute to that condition. According to the data, the highest prevalence of dysphagia is observed among individuals aged 65 and older, excluding patients with stroke, neurodegenerative disorders, head and neck cancer, and other conditions [1].

Dysphagia management is critical since difficulty swallowing can lead to malnutrition and dehydration, aspiration pneumonia, poor general health, chronic lung disease, choking, and even death. According to one’s study, malnutrition induces dysphagia, and conversely, dysphagia can also contribute to malnutrition. Approximately 3% to 29% of patients suffer from malnutrition and dysphagia [2]. One of the interventions advised by speech therapists and dietitians is using a texture-modified diet (TMD) to ensure that patients can eat by themselves and consume adequate daily nutrients to avoid malnutrition and support healing.

In healthcare food service, food handlers are responsible for preparing TMD for a patient with dysphagia. There are three types of TMD served in Malaysian hospitals: blended diet, minced diet, and mixed porridge diet—all based on the Ministry of Health’s diet manual. These textures are prescribed for various reasons, such as post-surgery dentures, one of which includes dysphagia. Therefore, the content is very general and not specific to the needs and requirements of those with dysphagia. The texture of the dysphagia diet must be technically accurate in preparation, nutrient-dense, and tasty in order to support bolus preparation in the mastication stage of digestion. According to a recent survey, difficulties swallowing solid foods occupied the largest percentage (29.5 percent), followed by coughing while/after swallowing (28.6 percent). Thus, it is critical to ensure that the texture of the diet is appropriate according to the patient’s ability to swallow [3].

Based on the International Dysphagia Diet Standard Initiatives framework (IDDSI), an 8-level continuum (0–7) classifies food and beverages. Beverages are measured from Levels 0–4, while meals are measured from Levels 3–7. After examination by a speech therapist, patients are assigned the appropriate diet texture level according to the IDDSI framework. Despite the IDDSI framework being adopted by the majority of nations across the world, Malaysia has yet to join the fray, and practices differ from one hospital to the next. The current practice in government hospitals is based on the Hospital Diet Manual. There is no specific texture modification or consistent diet for patients with dysphagia. However, it has been categorized into general terms such as blended diet, mince diet, and mixed porridge. The nomenclature used to describe each TMD diet and the texture assessment method is not standardized [4,5].

The pureed diet was chosen for this study to assess food handlers’ knowledge, attitude, and self-reported practices when preparing pureed diets for dysphagia therapy. In this regard, the pureed diet, which is at level 4 of the IDDSI framework, is expected to be soft, moist, and keep its form when served to the patient [6]. A pureed diet is characterized by two categories: structure and lubrication. The degree of structure is a combination of the bolus’ textural and particulate qualities, including hardness, cohesiveness, adhesiveness, and particle size. In contrast, the degree of lubrication is determined by surface properties and the feeling of wetness and juiciness of the bolus, such as moisture content and slipperiness [7].

According to research on plate waste, pureed diets amounted to up to 65 percent of total TMD’s plate waste in one of the hospitals in Malaysia [8]. This was contributed by several factors, including the texture itself. The mechanical alteration makes the pureed meal appear less appetizing because it resembles baby food. Thus, the pureed meal may not meet the patient’s preference in terms of texture and flavor, resulting in significant plate waste and malnutrition among dysphagia patients [8,9].

A pureed diet is usually deemed to be less palatable. Despite the fact that food texture is necessary for a total sensory experience, when it comes to dysphagia, the safety of the patient during food intake is the primary issue to focus on. As a result, food handlers must have high knowledge, maintain a positive attitude, and engage in regular practice when preparing pureed diets. This ensures that the standard for pureed food is met, promoting the safety of the patients. Hence, food handlers must be properly trained, informed, and have good practice in preparing pureed diet, which is very technical. In addition, the existing manuals for food handler instruction in government hospitals are particularly generic in their approach to TMD. Additionally, they do not focus in depth on each kind of TMD, particularly pureed diets. Presently, there is no research on knowledge, attitude, and practice (KAP) among food handlers toward TMD preparation in healthcare food service. Thus, the aim of this study is to assess the knowledge, attitude, and practice (KAP DYS-Puree) towards pureed diet preparation among food handlers for dysphagia management.

## 2. Materials and Methods

### 2.1. Data Collection

This study has been approved by the Medical Research and Ethics Committee (MREC), the Ministry of Health (NMRR-19-2804-47794 IIR), and the Universiti Kebangsaan Malaysia Medical Research Ethics Committee (UKM PPI/111/8/JEP-2019-682). The questionnaire was distributed to food handlers from three selected government hospitals: Kuala Lumpur Hospital (HKL), located in the capital city of Kuala Lumpur; Tuanku Ampuan Rahimah Hospital, Klang (HTAR), located in the western part of the state of Selangor; and Serdang Hospital, located in the southern part of Selangor. These hospitals were chosen as they were among the largest hospitals on the west coast of peninsular Malaysia, with many food handlers in their operation. Data collection focuses on the socio-demographic characteristics of food handlers, including their location in the hospital, gender, age, education level, and position within the organization. Additionally, they are asked whether they have received basic food handler training certification from a Ministry of Health-accredited trainer, which is mandatory for all food handlers in Malaysia, and whether they have received a typhoid injection. In terms of foodservice provision, Serdang Hospital has outsourced its operation to a private caterer, while HKL and HTAR run an in-house operation. The sample size calculation using the formula by Krejcie and Morgan determined the need for 161 respondents for validation tests with a significant level and degree of accuracy of 5%. Prior to data collection, all food handlers were informed about the study’s aims, and written consent was obtained. Hospitals were selected based on patient bed counts and food serving systems (in-house or outsourced). Data collection was implemented online and led by the catering officer of each hospital, who was assigned as an enumerator. Written consent forms were distributed to food handlers who were directly involved in preparing pureed diets before data collection, and they were to be returned to the catering officer in charge. Consenting food handlers were then provided a link to complete the digital questionnaire using Google Forms after working hours, so they could not influence each other’s responses.

### 2.2. The Questionnaire

The KAP DYS-Puree questionnaire was a newly developed questionnaire consisting of 40 items querying knowledge, attitude, and practice related to pureed diet preparation with selected domains [4] based on previously published studies [10,11,12]. Despite the fact that the questionnaire has been validated through face validation, content validation, content reliability, and construct validation and has undergone test–retest on a subsample of food handlers in hospitals, this instrument was employed for the first time for this study. It was developed in Bahasa Malaysia (Malay) with five sections. Section A of KAP DYS-Puree was designed to collect information on socio-demographics. Section B included six questions about food handler certificates and typhoid vaccination, a pre-requisite for all food handlers, and current practices in preparing a pureed diet in their organization. Section C (knowledge) consisted of 20 polytomous questions (Yes, No, and Not Sure) covering four knowledge domains: dysphagia, puree diet preparation, food safety, and nutrition. Food handlers who answered all 20 questions on knowledge would be scored 1 if they answered correctly and 0 if they were incorrect or not sure. The total percentage they scored will be categorized to fit into a modified Bloom’s cut-off point, with the scores of the proportions ranging between 80 and 100 percent for good (>17 points), moderate between 50 and 79 percent (11–16 points), and poor below 50 percent (<10 points) [12]. Section D (attitude) and Section E (practice) each had ten items on a five-point Likert scale (1 = disagree strongly; 2 = disagree; 3 = neutral; 4 = agree; and 5 = agree strongly). Each of the questions is themed based on its original questionnaire in order to maintain consistency and coherence throughout the survey.

## 3. Results

### 3.1. Socio-Demographic Characteristics

This study recruited 161 food handlers, the majority of whom were from Hospital Kuala Lumpur (53.4%), Malaysia’s largest hospital, followed by Hospital Tuanku Ampuan Rahimah Klang (29.8%), and Hospital Serdang (16.8%). Table 1 summarizes the demographic characteristics of the participants, with 115 (71.4%) females and 46 (28.6%) males taking part in the study. The majority of food handlers were between the ages of 30–39 years old (51.6%). Malays were the most common race in the research, accounting for 157 (97.5%) of them since a high percentage of Malay individuals work in the government’s healthcare foodservice sector. The data revealed that the majority of food handlers are educated, with 30.4% of them being diploma holders, followed by 21.1% holding basic culinary certificates. As the aim of the study was to assess the KAP among food handlers on puree diet preparation, food handlers in the position of Catering Assistant grade N19 (60.95%) were recruited the most, followed by Assistant Catering Officer grade C29 (24.2%).

Based on the scoring marks from all 20 questions answered by all food handlers, Table 2 summarizes the fractions of food handlers’ score percentages on each item included in the questionnaire. In total, 4 food handlers (2.5%) have marks below 25%, 46 food handlers (28.6%) scored below 50%, 110 food handlers (68.3%) attained below 75%, and 1 food handler managed to get higher than the 75% scoring. Knowledge of pureed diet preparation domains showed low to moderate score percentages (M = 57.54, SD = 12.33). Thus, to sum up the findings from Table 2, it is clear that the knowledge level among handlers was low to moderate towards pureed diet preparation.

### 3.2. Attitude

As displayed in Table 3, the scores for attitudes were moderate (3.24 ± 1.35) for work compliance, indicating that food handlers have a positive attitude about following the management’s guidelines and standards. Item 6, which covers self-reactiveness, had a low score, followed by 7 items with a moderate score ranging from 2.68 to 3.61 mean value. Items 4 and 8 showed high scores in recipe compliance and attitude towards resources.

### 3.3. Practice

The practice contained 10 variables about the practice of food handlers toward pureed diet preparation. The mean percentage of food handlers’ scores, mean, and standard deviation are presented in Table 4. The mean scores for all 10 items in the food practice among food handler’s variables range from 2.25 to 3.97, showing a moderate practice towards pureed diet preparation. Items 7 and 10 showed high mean values, followed by six items with moderate values and two with low practice values: pre-preparation and tools equipment.

## 4. Discussion

This study aimed to assess the knowledge, attitude, and practice towards pureed diet preparation among food handlers. As the questionnaire has been validated and is reliable for dysphagia management, the study revealed that most food handlers were not equipped with sufficient knowledge of pureed diet preparation specifically for dysphagia management. The findings were concerning as patients with dysphagia are prone to aspirate due to wrong food texture. It is clear that the level of knowledge among food handlers for the preparation of a pureed diet for dysphagia patients was below par since preparations are very technical to achieve the right texture in order to be safe for consumption.

The findings show that food handlers had the lowest knowledge of diet preparation, which is in line with the low-score results for that attribute. It correlates with the findings obtained from attitude and practice, variables that are directly related to responses on preparation, such as “I blend all ingredients before cooking the pureed diet”, “I puree foods using only plain water that has been boiled” and “I use a special blender for the puree diet”. This finding directly concerns the texture of the puree diet, which should be emphasized in the preparation procedures. Since the previous study has shown that knowledge has a strong association with attitude and practice, these findings indicate that low to moderate knowledge, with moderate attitude and practice, will increase the risk of aspiration in patients with dysphagia. This is due to the unsuitability of texture because of poor information and incorrect conceptions about dysphagia that contribute to noncompliance [9]. This lack of conformity in attitude and practice can be fatal, as a recent study found that food asphyxiation was the cause of death in 14 of 1087 (1.3 percent) autopsies conducted over a 5-year period [13]. Although the study is not directly connected to pureed diet preparation, the referenced study gives persuasive evidence of the possible implications of poor food handling practices. By relating this information to the specific context of pureed meal preparation, it emphasizes the need for food handlers to prioritize correct skills and adherence to guidelines to reduce the risk of food aspiration for people with swallowing difficulties.

As a result, the citation serves as a reminder of the need to follow proper procedures while preparing pureed diets to protect the safety and well-being of individuals who rely on such diets. It emphasizes the possible severity of repercussions when necessary processes are not followed and the importance of adhering strictly to standards to avoid accidents such as food poisoning.

As the most concerning results for attitude and practice are towards preparing pureed diets, management needs to strengthen the ongoing training with in-depth content focusing on pureed diets. An extensive module is also suggested to aid food handlers in achieving good practice in pureed diet preparation. Previous findings revealed that training courses, extensive experience, and working in government hospitals all have a substantial impact on achieving better knowledge ratings [14]. Even though most food handlers were qualified with tertiary education, a weak association was observed. This could be due to poor knowledge of pureed diet preparation from a lack of training, or if training is provided, it is not focused enough on the TMD. It is recommended that comprehensive scheduled training programs for food handlers be given to help them increase their knowledge since poor preparation knowledge leads to weak practices.

Next, the existing reference used by food handlers, which is the Hospital Diet Manual 2016, from which food handlers were disengaged in terms of the process management of pureed diet preparation, was concerning. Thus, it is recommended that a comprehensive pureed diet module be provided so food handlers can have a better idea and understanding of how the operation and process management fit into the larger picture of the organization to guide them in dysphagia management. The outcome of this study feeds the needs of food service operations by emphasizing training through a comprehensive module for food handlers’ guidance.

In addition, hands-on training and work rotation should be considered by the management to let all food handlers experience achieving a certain number of hours in the preparation of pureed diet. Some practices observed that only a certain number of food handlers were assigned to the TMD preparation, which needs to be addressed by the management. The findings are in line with the previous study, which found that higher levels of education, food handling experience, and employment position do not produce a better outcome [6]. This is good practice, as food handlers are rotated, and the menus are cycled accordingly. Thus, it is crucial to have orders from one source rather than multi-orders from different food handlers and supervisors to avoid mistakes during preparation.

The results of the previous study also revealed that a positive attitude among food handlers toward following the standard practice of the food operation inversely affected the standard preparation time. Food handlers tend to complete their work as fast as they can, which affects the safety of the food. As food is prepared in a short time, the holding time becomes longer. It can lead to microbial growth if the food is not held at the right temperature, especially in hospitals such as HKL and HTAR, which use decentralized food service systems whereby food is distributed to most wards in bulk. Pureed food cools much faster after it is cooked due to its small surface area.

Thus, this area is yet to be explored and understood to meet the needs of the operation while maintaining the safety of the pureed diet prepared. Even though the food handlers have offered positive responses, in reality, they may not practice them when handling foods [15]. This result is in line with the previous study that showed the majority of individuals who had a good attitude tended to have a low to moderate knowledge level [16].

One of the major highlights from the result shows that attitudes towards pureed diet preparation lean towards adjusting the recipe, and there is a tendency not to follow any references. This finding showed a negative attitude, especially toward pureed diet preparation, as the accuracy of the texture depends on standard operating procedures during food preparation. Pureed diets cannot be treated as cooking a normal diet, where the arts of food are maneuvered to make it tasteful. However, a pureed diet is more focused on safety, followed by palatability.

On the other hand, the result also discloses the attitude among food handlers toward getting the job right, even if it is not something they enjoy doing. This positive attitude is crucial among food handlers, especially those involved in texture modification diets, as the preparation is more complicated when compared to normal diet preparation since it involves patient safety. However, management has to step up to underpin all issues related to pureed diet preparation, as previous studies show that even though respondents had a positive attitude, they had low knowledge and practice [17]. Thus, it is crucial to have high knowledge, attitude, and practice as abovementioned in pureed diet preparation to aid the patient’s healing process through foods.

## 5. Conclusions

In conclusion, it is recommended that all food handlers participate in intensive training to enhance their knowledge of pureed diet preparation. The outcome of this study has shown the loopholes that can be worked on to strengthen the knowledge among food handlers and lead to a good attitude and practice toward pureed diet preparation. Comprehensive training is a must, and it should be aligned with a procedure that is being monitored to verify its efficacy and continuous improvement. This alignment enables a continual feedback loop in which the training program is customized based on monitoring outcomes. The limitation of the study is that there is no dedicated comprehensive reference for TMD preparation and no IDDSI compliance since IDDSI is yet to be implemented in Malaysia. In summary, efforts to provide a main reference guide for food handlers to use and work with need to be amplified. This will lead to a significant improvement in food handlers’ attitudes and practices, and the safety of dysphagia patients can be maintained. 

## Figures and Tables

**Table 1 healthcare-11-02026-t001:** Socio-demographic characteristics of the study population, n = 161.

Hospital	(n)	Percentage (%)
Hospital Kuala Lumpur	86	53.4
Hospital Serdang	27	16.8
Hospital Tuanku Ampuan Rahimah Klang	48	29.8
Gender		
Male	46	28.6
Female	115	71.4
Age		
18 to 29 years old	35	21.7
30 to 39 years old	83	51.6
40 to 49 years old	21	13.0
50 to 60 years old	22	13.7
Race		
Malay	157	97.5
Indian	4	2.5
Education		
Master’s Degree	4	2.5
Bachelor’s Degree	8	5.0
Diploma	49	30.4
Basic Certificate in Culinary	34	21.1
High Secondary	43	26.7
Lower Secondary	18	11.2
Position		
Catering Assistant grade N1	11	6.8
Catering Assistant grade N19	98	60.9
Assistant Catering Officer grade C29	39	24.2
Assistant Catering Officer grade C32	8	5.0
Catering Officer grade C41	5	3.1
Food Handler’s Training Certificate		
Yes	156	97
No	5	3
Typhoid Injection Certificate		
Yes	140	87
No	21	13

**Table 2 healthcare-11-02026-t002:** Knowledge score based on domains.

Domain	Item	N	Score Yes (%)	Score No and Not Sure
**General Knowledge of Dysphagia**	Item 1: General knowledge of dysphagia diet	161	98.8	1.2
	Item 2: General knowledge of dysphagia condition	161	70.2	29.8
	Item 3: General knowledge of texture modification type of dysphagia food	161	96.9	3.1
	Item 4: General knowledge of dysphagia diet preparation	161	20.5	79.5
	Item 5: General knowledge of types of the dysphagia diet	161	99.4	0.6
**Knowledge of Pureed Diet Preparation**	Item 1: Preparation of pureed diet	161	39.8	60.2
	Item 2: Ingredients of pureed diet for dysphagia	161	31.1	68.9
	Item 3: Types of vegetables for pureed diet	161	94.4	5.6
	Item 4: Method of cooking for pureed diet	161	5.6	94.4
	Item 5: Method of preparation for suitable texture in pureed diet	161	67.7	32.3
**Knowledge of Food Safety**	Item 1: General knowledge of temperature for pureed diet safety	161	100	0
	Item 2: Holding time of pureed diet	161	64.6	35.4
	Item 3: Temperature of cooking	161	65.2	35.4
	Item 4: Utensil and tools safety during pureed diet preparation	161	77.0	23
	Item 5: Holding temperature for pureed diet	161	23.6	76.4
**Knowledge on Nutrition**	Item 1: Vegetable selection for pureed diet preparation	161	72.0	28
	Item 2: Effect of temperature on nutrients	161	80.1	19.9
	Item 3: Fats for pureed diet	161	50.3	49.7
	Item4: Food pyramid selection	161	77.6	22.4
	Item 5: Calorie of a pureed diet	161	9.9	90.1

**Table 3 healthcare-11-02026-t003:** Food handler’s attitude towards pureed diet preparation.

		Disagree Strongly (N%)	Disagree (N%)	Neutral (N%)	Agree (N%)	Agree Strongly (N%)	Mean and Standard Deviation
**1.**	Item 1: Attitude toward texture modification food preparation	35 (21.7)	30 (18.6)	42 (26.1)	17 (10.6)	37 (23.0)	2.94 ± 1.47
**2.**	Item 2: Attitude toward accuracy in food preparation	39 (24.2)	11 (6.8)	20 (12.4)	27 (16.8)	64 (39.8)	3.41 ± 1.63
**3.**	Item 3: Attitude toward the suitability of the ingredients	51 (31.7)	18 (11.2)	43 (26.7)	29 (18.0)	20 (12.4)	2.68 ± 1.40
**4.**	Item 4: Attitude toward standard recipe compliant	13 (8.1)	3 (1.9)	6 (3.7)	22 (13.7)	117 (72.7)	4.41 ± 1.19
**5.**	Item 5: Attitude towards self-meticulousness	13 (8.1)	12 (7.5)	33 (20.5)	26 (16.1)	77 (47.8)	3.38 ± 1.31
**6.**	Item 6: Attitude toward self-proactiveness	91 (56.5)	10 (6.2)	18 (11.2)	19 (11.8)	23 (14.3)	2.21 ± 1.55
**7.**	Item 7: Attitude toward obedience	21 (13.0)	16 (9.9)	69 (42.9)	24 (14.9)	31 (19.3)	3.17 ± 1.23
**8.**	Item 8: Attitude toward resources	15 (9.3)	10 (6.2)	36 (22.4)	30 (18.6)	70 (43.5)	3.81 ± 1.31
**9.**	Item 9: Attitude toward time urgency	61 (37.9)	15 (9.3)	15 (19.3)	33 (20.5)	37 (23.0)	2.81 ± 1.69
**10.**	Item 10: Attitude toward self-skill	36 (22.4)	15 (9.3)	43 (26.7)	24 (14.9)	43 (26.7)	3.14 ± 1.48

**Table 4 healthcare-11-02026-t004:** Food handler’s practice towards pureed diet preparation.

		Never (N%)	Seldom (N%)	Sometimes (N%)	Frequent (N%)	Always (N%)	Mean and Standard Deviation
**1.**	Item 1: Accuracy	18 (11.2)	40 (24.8)	55 (34.2)	30 (18.6)	18 (11.2)	2.94 ± 1.15
**2.**	Item 2: Pre-preparation	67 (41.6)	26 (16.1)	40 (24.8)	13 (8.1)	15 (9.3)	2.27 ±1.33
**3.**	Item 3: Standard recipe	30 (18.6)	23 (14.3)	27 (16.8)	27 (16.8)	54 (33.5)	3.32 ± 1.52
**4.**	Item 4: Tools and equipment	63 (39.1)	33 (20.5)	38 (23.6)	16 (9.9)	11 (6.8)	2.25 ± 1.26
**5.**	Item 5: Teamwork	12 (7.5)	21 (13.0)	43 (26.7)	33 (20.5)	52 (32.3)	3.57 ± 1.27
**6.**	Item 6: Operational work flow	25 (15.5)	19 (11.8)	40 (24.8)	33 (20.5)	44 (27.3)	3.32 ± 1.40
**7.**	Item 7: Time Management	17 (10.6)	9 (5.6)	19 (11.8)	33 (20.5)	83 (51.6)	3.97 ± 1.39
**8.**	Item 8: Competency	22 (13.7)	18 (11.2)	43 (26.7)	35 (21.7)	43 (26.7)	3.42 ± 1.57
**9.**	Item 9: Cooking Plan	22 (13.7)	18 (11.2)	43 (26.7)	35 (21.7)	43 (26.7)	3.37 ± 1.35
**10.**	Item 10: Standard operating procedure practices	12 (7.5)	8 (5.0)	32 (19.9)	32 (19.9)	77 (47.8)	3.96 ± 1.25

## Data Availability

The data provided in this analysis are part of the continuing doctoral program in A.A.A. analysis. We were also unable to disclose the data publicly. However, it is accessible from the respective author upon request (N.H.R.).

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
