# Peer review of "An Assessment of Knowledge, Attitudes and Practices on Pureed Diet Preparation (KAP DYS-PUREE) among Food Handlers in Malaysian Hospitals for Dysphagia Management"

_healthcare, 2023, doi:10.3390/healthcare11142026_

Round 1
Reviewer 1 Report
The paper is interesting as nutrition in healthcare is very relevant. However, some pieces of the manuscript are not clear to me.
- Text
3.1 Sociodemographic characteristics
Table 1
Are the education titles and the position in the profession ordered in a descending way? Please explain.
The cutoffs delimiting low, moderate, and high are not reported, so it is difficult to understand the results.
How was the score percentage calculated?
What do M and SD correspond to? Is it the mean of the score percentage?
Is the M the arithmetic mean? Please, the first time, use the extended name.
The figures are reported only here, while these are not included in tables.
Table 2. There are some figures out from the rows around the "Knowledge of Food Safety" box.
3.2 Attitude
The range in Table 2 is 2.68-3.81 and not 3.62 (line175).
Discussion
Lines 235-236 "Thus, it is crucial to have orders from one source rather than multi orders from different food handlers and supervisors to avoid mistakes during preparation"
What does "multi orders" mean ?
Lines 243-244 "especially in hospital like HKL and HTAR that are using decentralized food service systems".
This is different from what the Authors wrote in 106-107 lines, where "Serdang Hospital has its operation to a private caterer, while HKL and HTAR both run an in-house operation".
- Other questions
How are food handlers recruited in the hospitals?
Do the Authors think that also monitoring the quality of the meals and using eventually tasters could help enhance the nutrition service in the hospital?
- Minor editing
The enumeration of the references does not seem correct, as the first number is 13 and should be 1.
I would reverse the order of The Questionnaire and Data collection as extensive descriptions are currently in 2.2 and then repeated in 2.1.
Reviewer 2 Report
It is a meaningful task to investigate the characteristics of food preparation experts who play an important role in the management of patients with dysphagia.
Page 3, Data collection, fifth line: ‘Malaysia’ repeated twice. Is it a typo?
One of the most important characteristics to be considered in meals provided to patients with dysphagia is the appropriate 'viscosity' for each patient. However, it seems that there is not much discussion about this in this paper.
Additional comments:
It is difficult to confirm the exact contents as the actual questions of the questionnaire used in this study are not presented.
There is no explanation on how to check and measure the viscosity of prepared food in the knowledge of Table 2 and the practice of Table 4. Even if pureed food is made with the same ingredients, the viscosity can vary greatly depending on various conditions. Therefore, in the method part, it is necessary to explain how the measurement was performed and to consider in the discussion part.
A pureed diet and a puree diet are being used interchangeably. You need to unify in one term coherently.
A survey was conducted with 161 food handlers working in 3 hospitals. In general, in order to get a picture of the current state of a particular problem, research is conducted with people working in as many organizations as possible. If a survey is conducted by multiple respondents working in the same hospital, similar results are likely to be obtained from those with similar demographic characteristics. Despite these risks, a justification must be given for conducting a survey of a large number of people working in a very limited hospital.
Author Response
Please see the attachement

Reviewer 3 Report
In their manuscript "An assessment of knowledge, attitudes and practices on pureed diet preparation (KAP DYS-PUREE) among Food Handlers in Malaysian Hospitals for Dysphagia Management" the authors present the results of a study that was conducted in 3 hospitals in Malaysia with the objective of evaluating the knowledge, the attitudes practices regarding the preparation of meals for dysphagic patients.
The argument is interesting, but the manuscript must be presented better.
The methodology is not well explained and that prevents a good understanding of the tables in the results which, among other things, are excessively summarized in the text. There is much more to say when commenting tables. The questionnaires used must be better explained and there is a lack of explanations regarding the statistical elaborations.
As an example I suggest you an article that uses the same methodology to detect the knowledge, attitudes and practices regarding Covid-19: https://bmcpublichealth.biomedcentral.com/articles/10.1186/s12889-021-10285-y/tables/2
Data collection: it must be explained n chronological order; from the moment of approval of the study, to the identification of the hospitals and calculation of the number of participants, the signing of the consents...
The questionnaire: it is not clear if it is a validated questionnaire, if it is so by whom and with what methods and results. Also, the questions need to be explained much better, as suggested by the example of the article I mentioned.
The presentation of the results in the tables: I think the % are enough without using the averages and the standard deviation which, if you decide to leave them, must be explained.
Furthermore, it would be advisable to try to correlate the main results to socio-demographic and education-related variables.
The discussion needs to be reorganized; in the introductory part, the objectives of the study and the main results must be explained with 2-3 sentences. Critical comments and miscellaneous observations based on comparisons with existing literature should follow.
In the final phase of the conclusions, the limits of the study must be exposed, which are not only those relating to the IDDSI scale. Likewise, it should be specified in more detail how the results of the study can be useful and how to use them.
English must be absolutely checked and corrected by a native speaker.
Must be improved!
Round 2
Reviewer 3 Report
the bibliographical references must all be arranged in order - the first reference number is 14!!! (line 42). that must be #1
line 22 - written like this, it appears that you prepared a questionnaire and then validated it on this occasion. in this case the results of the study would concern the validation and not the results of the study carried out with an already validated questionnaire. so i suggest "a validated questionnaire was used..."
lines 38-41- the sentence is incomprehensible. perhaps you intended to say "data on the prevalence of dysphagia, i.e. swallowing problems, vary from country to country and according to age... the highest prevalence is normally found in subjects aged 65 and over...
lines 45-49: wrong! the evidence, which is scarce and week, shows that the first treatment of dysphagia must be done with a modified diet but that only the supplemented diet has proven to be efficient in some cases in preventing malnutrition. Furthermore, it is not true that all patients can consume the modified diet alone because this diet is also prescribed for non-independent subjects (advanced dementia and similar). Why is the #1 reference about diabetes? There are more appropriate reviews and meta analysis on texture modified diets to quote.
line 57 - take off
comment bp9 - i confirm the comment
I suggest that you first describe IDDSI (lines 63-69) and then tell how diets are classified in hospitals of Malaysia (lines 50-61).
line 53 - for which conditions are they prescribed besides dysphagia?
the iddsi provides an 8 level continuum.....levels 3-7, and patients are provided with the appropriate diet texture level after being examined by a speech therapist.
line 74 - pureed diet is characterized
line 80 - worldwide or Malaysia?
bp10 and bp11: I agree with the comments
lines 243-245 the meaning is not clear
line 255: results of previous study
line 263: operation? It is not clear
page 2: there is a mess with page numbers, a part of discussion is indicated as page 2 of 12!!!!
lines 123-128 repeated, you already said that
methodology: it is still not clear if the questionnaire has been previously prepared and validated by others, or used for the first time (and therefore not validated). looking at the bibliography i would say that the correct answer is the first, but this needs to be explained better.
line 174: for attitudes were moderate (without shown to be)
line 177: had, not was found to have
comment bp14: i agree with comment
row 194: aimed, not aims
following sentences start both with "as" which should be avoided
Minor corrections are needed
